# Improved Photoluminescence Performance of Eu^3+^-Doped Y_2_(MoO_4_)_3_ Red-Emitting Phosphor via Orderly Arrangement of the Crystal Lattice

**DOI:** 10.3390/molecules28031014

**Published:** 2023-01-19

**Authors:** Fan Chen, Muhammad Nadeem Akram, Xuyuan Chen

**Affiliations:** Department of Microsystems, Faculty of Technology, Natural Sciences and Maritime Sciences, Campus Vestfold, University of South-Eastern Norway, 3184 Borre, Norway

**Keywords:** photoluminescence, red emitting, down conversion, thermal quenching, Eu^3+^-doped, lattice order, laser lighting

## Abstract

In this study, we developed a technology for broadening the 465 nm and 535 nm excitation peaks of Eu^3+^:Y_2_(MoO_4_)_3_ via crystal lattice orderly arrangement. This was achieved by powder particle aggregation and diffusion at a high temperature to form a ceramic structure. The powdered Eu^3+^:Y_2_(MoO_4_)_3_ was synthesized using the combination of a sol–gel process and the high-temperature solid-state reaction method, and it then became ceramic via a sintering process. Compared with the Eu^3+^:Y_2_(MoO_4_)_3_ powder, the full width at half maximum (FWHM) of the excitation peak of the ceramic was broadened by two- to three-fold. In addition, the absorption efficiency of the ceramic was increased from 15% to 70%, while the internal quantum efficiency reduced slightly from 95% to 90%, and the external quantum efficiency was enhanced from 20% to 61%. More interestingly, the Eu^3+^:Y_2_(MoO_4_)_3_ ceramic material showed little thermal quenching below a temperature of 473 K, making it useful for high-lumen output operating at a high temperature.

## 1. Introduction

Phosphor-converted white-light-emitting diodes (pc-WLEDs) [1,2] have become prominent in next-generation solid-state lighting [3,4]. A conventional pc-WLED is composed of a blue LED and yellow phosphor material, which emits cold white light due to lack of a red component in the emission spectrum. The addition of red-emitting phosphor material can complement the red component of the emission spectrum, thereby improving the overall lighting quality. Hence, looking for a high-quality, inexpensive, red-emitting phosphor material is crucial to the development of next-generation lighting technology for solid-state lighting. Unfortunately, it is very hard to find a red-emitting phosphor material that provides all the advantages simultaneously, such as low thermal quenching, high quantum efficiency, broadband near-red emission, high saturation power density, and strong excitation at typical blue excitation wavelengths. Therefore, exploiting high-quality red-emitting phosphor materials remains challenging [5].

Currently, potential red-emitting phosphor materials used in the pc-WLED field are mainly Mn^4+^-doped fluoride [6,7,8], Eu^2+^-doped nitride [9,10] and Eu^3+^-doped inorganic material [11,12]. Mn^4+^-doped fluoride red-emitting phosphor material has been widely favored due to its low cost (no rare-earth elements) and high quantum efficiency [13]. However, unsatisfactory thermal stability and very low saturation power density severely restricts its applications. In addition, during the Mn^4+^-doped fluoride synthesis process, the extensive use of hydrofluoric acid is not safe for the environment. The most widely used commercially available red phosphors are Eu^2+^-doped nitride phosphors, such as CaAlSiN_3_:Eu^2+^ [14] and Sr_2_Si_5_N_8_:Eu^2+^ [15]. Although these phosphors realize high quantum efficiency, the harsh synthesis conditions (high temperature and high pressure) and deep-red emission makes it difficult for them to become ideal candidates for pc-WLED. The trivalent europium cation (Eu^3+^) is well known for its strong luminescence in the red spectral region due to the ^5^D_0_→^7^F_J_ electron transition (J = 0–4) from the ^5^D_0_ excited state to the J levels of the ground state ^7^F. The oxide hosts (molybdate, tungstate, niobate, and tantalate) are favored for Eu^3+^-doped red-emitting phosphor materials due to their excellent luminescent properties, such as high quantum efficiency, high color purity, and high thermal stability. The photoluminescence properties of Eu^3+^-doped inorganic materials belong to the narrow-band type [11]; the excitation and emission spectra are both narrow. In pc-WLED applications, the narrow-band emission spectrum of phosphor materials is advantageous as it can provide high color purity. On the contrary, narrow-band excitation can easily cause mismatching with the excitation source wavelength due to the wavelength shift of the excitation source operating at an elevated temperature. In practical applications, the wavelength of the LED/LD sources will shift to the longer wavelength at high temperatures that can result from the heat accumulation over a longer working period [16]. Therefore, Eu^3+^-doped inorganic red-emitting phosphor materials could be very competitive candidates for pc-WLED application if their excitation waveband could be broadened to cover the wavelength shift range of the excitation source operating over a long period of time [17,18]. Determining how to broaden the excitation waveband without affecting the other luminescent properties of Eu^3+^-doped inorganic red-emitting phosphor remains an unsolved problem.

In this study, we focused on developing high-performance Eu^3+^:Y_2_(MoO_4_)_3_ inorganic red-emitting material [19,20,21,22]. The energy-level structure of Eu^3+^ embedded in a medium is formed by several perturbations acting on the Eu^3+^ ion, including electron repulsion, spin–orbit coupling, and crystal field perturbation [11,23]. Together, the electron repulsion and spin–orbit coupling decide the Eu^3+^ energy levels of ^7^F_J_ and ^5^D_J_ that basically determine the excellent photoluminescence performance of Eu^3+^. The crystal field perturbation has less of an impact on the overall energy level structure of Eu^3+^, but causes subtle adjustment in the sub-level splitting. Using a high-temperature sintering process, we modified the orderly arrangement of the lattice with the aim of the crystal field perturbation promoting the splitting of the 7F and 5D energy levels, which is reflected in the spectrum as the broadening of the excitation peaks.

## 2. Results and Discussion

### 2.1. Microscopic Morphology Study

The surface microscopic morphology of the powder and ceramic samples was analyzed by SEM, as shown in Figure 1a–d, corresponding to the powder, 900C, 1000C, and 1100C samples, respectively. The microstructure of the powder sample had irregularly shaped, tiny dispersed pieces with a size of approximately 100 nm. For the three ceramic samples, particle aggregates were clearly observed. The size of the particles was significantly different depending on the calcinating temperature. The average size of the particles increased from 3 µm to 25 µm when the calcinating temperature increased from 90 °C to 1100 °C. Powder samples were made into dense cakes for the high-temperature calcination process, which is conducive to the mutual diffusion of small particles of the powder at a high temperature to cluster into larger particles. The higher-temperature process leads to a faster diffusion speed, a greater diffusion range, and a higher active energy for crystallizing larger particle sizes.

The chemical composition of the samples was confirmed using the energy-dispersive spectrometry (EDS) measurement, as shown in Figure 1e,f, corresponding to the powder sample and the three ceramic samples. The element tables inserted in Figure 1e,f show that the powder sample and ceramic samples had the same composition, considering the measurement error, and the content of each element was in total agreement with the stoichiometry value. Using EDS analysis data, we could distinguish the main elements in the sample and approximate the proportion of each element, so that we could find no impure crystalline contributions. Therefore, it can be concluded that we successfully synthesized the expected Eu^3+^-doped Y_2_(MoO_4_)_3_ red phosphor.

### 2.2. Crystal Structure

Trivalent rare-earth (RE) molybdates of the RE_2_(MoO_4_)_3_ family can form four different crystal structures, including monoclinic (C2/c), orthorhombic (Pba2, Pbna), and tetragonal (P$\bar{4}$2_1_m), due to the different cations and annealing temperature [22,24,25]. The XRD results of the powder and the ceramic samples are shown in Figure 2a, which presents a tetragonal structure belonging to space group P$\bar{4}$2_1_m (Y_2_(MoO_4_)_3_ COD-7054114, identified from the Crystallography Open Database [21], as shown in Figure 2b. In the tetragonal Y_2_(MoO_4_)_3_, each unit cell included two asymmetric units, four octahedral [YO_7_] groups (an oxygen atom is outside the unit cell), four internal tetrahedral [MoO_4_] groups, eight corner-sharing [MoO_4_] groups, and two face-sharing [MoO_4_] groups. The structures of the octahedral [YO_7_] group and the tetrahedral [MoO_4_] group are shown in Figure 2c,d. Each octahedral [YO_7_] group is connected to six tetrahedral [MoO_4_] groups, and one [YO_7_] group that belongs to another unit cell, and each [MoO_4_] group is connected to four [YO_7_] groups. The tetragonal (P$\bar{4}$2_1_m) crystal lattice constants are a = b = 7.46 Å, c = 10.754 Å, and α = β = γ = 90°.

The XRD diffraction pattern of the powder sample was in complete agreement with the standard card (COD-7054114), and no difference was observed. However, the XRD diffraction patterns of the three ceramic samples were slightly different from the standard card, with a strong diffraction peak between 62° and 64° compared with the weak diffraction peak in the powder sample, represented by the red dashed rectangle in Figure 2a. Repeated experiments proved that this was a systematic phenomenon, rather than an accidental coincidence. According to the previous EDS results, the elemental composition and content were the same, and we can confirm that this new diffraction peak was not from impurities contaminating the ceramic samples during calcination. It is obvious that the enhanced diffraction peak at 62°–64° in the diffractograms of the three ceramics was introduced by the ordered lattice.

### 2.3. UV–Vis Absorption Spectra

The ultraviolet–visible absorption spectra of the powder and the ceramic samples are shown in Figure 3a, which were composed of a strong broadband absorption and multiple-line-type absorption peaks. This strong broadband absorption was attributed to the host absorption band (ligand-to-metal charge transfer) and occurred in the ultraviolet range. The multiple-line-type absorption peaks were attributed to the direct-absorption Eu^3+^ (7F→5D electron transition) and were located from 380 nm to 600 nm [26,27]. Comparing the powder and the ceramic samples, the entire absorbance of the ceramic samples was much higher than for the powder sample, especially in the line-type absorption peaks; it was approximate three-fold higher (as shown in the inset of Figure 3a). It is clear that the ceramic samples had larger crystal grains and reduced grain boundaries, which ensured that enough luminescent centers received and absorbed the excitation light.

### 2.4. Optical Band Gap Analysis

The optical band gap of the sample host can be derived from the absorption spectrum near the band edge using the empirical Formula (1) [28]:(1)$$\alpha hv=A_{0}{\left(hv-E_{g}\right)}^{n}$$
where α is the absorbance, h is Planck’s constant, ν is the frequency, $A_{0}$ is a constant, $E_{g}$ is the optical band gap, and n is an exponent value, which equals ½, in this case. The modified empirical formula is given as the following:(2)$${\left(\alpha hv\right)}^{2}=A_{1}\left(hv-E_{g}\right)$$
where *A*_1_ is a constant. Figure 3b–e presents ${\left(\alpha hv\right)}^{2}$ versus hv, and the optical band gap of the hosts was estimated to be 3.52, 3.31, 3.21, and 3.28 eV by the intercepts of ${\left(\alpha hv\right)}^{2}=0$, which corresponds to the powder, 900C, 1000C, and 1100C samples, respectively. We can see that the optical band gaps of the ceramic samples were reduced compared to the powder. A reasonable explanation for this is that the crystal field effects of the ordered lattice were superimposed and enhanced, which in turn, affected the optical band gaps. By calculating the bandgap we can understand more about the absorption in the UV band of the host material, especially the energy transfer process of the absorption in the UV band for enhancing the red emission [29].

In summary, (1) the ceramic samples with a large-sized crystal grain enabled a high number of the luminescent centers to be active compared to the powder sample; therefore, the absorption intensity of the ceramic samples was much greater than that of the powder sample; and (2) the ordered arrangement of the lattice via calcination in the ceramic samples led to the growth of crystal grains that did not have an improved crystalline quality compared to that of the small crystal grains in the powder samples. Due to the high-level stress at the grain boundaries in the ceramic samples, there were more defects and strains in the extended part of the re-grown crystal grains. Therefore, the crystal field effect was enhanced.

### 2.5. Excitation and Emission Spectra

Excitation and emission spectra of the powder and the ceramic samples are shown in Figure 4a–d. The excitation spectra were measured at the emission wavelength of 616 nm, and the emission spectra were measured under the 465-nm excitation wavelength. As is well known, the emission peaks of Eu^3+^ located at 591, 616, 656, and 703 nm are attributed to ^5^D_0_→^7^F_J_ (J = 0–4) electron transitions. The emission spectra of the ceramic and powder samples were found to be in perfect agreement, which indicates that the crystal field effect had no effect on the ^5^D_0_ and 7F energy levels. The excitation peaks of Eu^3+^ located at 362, 382, 395, 416, 465, and 535 nm were attributed to electron transition from the ground state ^7^F_0_ to the excited state ^5^D_4_, ^5^L_7_, ^5^L_6_, ^5^D_3_, ^5^D_2_ and ^5^D_1_, respectively. In addition, a broad excitation band located in the ultraviolet region belongs to the charge transfer band (CTB) [30] and arises from two primary mechanisms [27,31,32]. The first is the charge transfer of Mo-O (electronic transfer from the 2p orbital of the O^2−^ ligand to the 5d orbital of metal Mo^6+^) in the [MoO_4_]^2−^ group of the host crystal. The second is the charge transfer of Eu-O (electronic transfer from the 2p orbital of O^2−^ to the 4f orbital of Eu^3+^ [19]) between the neighboring O^2−^ and Eu^3+^. These two CTBs are overlapped in the ultraviolet region due to the fact that the energy levels of the 5d orbital of the metal Mo^6+^ and the 4f orbital of Eu^3+^ are very close [33,34]. 

Compared with the powder sample, the excitation spectra of the ceramic samples had the following characteristics. First, an intense CTB broadband from Eu-O appeared in the ultraviolet region, which was illustrated by the fact that the ceramic sample could be effectively excited by 230–270 nm ultraviolet light. The reason for this is that the grain boundary of the ceramic sample was much less than that of the powder sample, and the reflection caused by the grain boundary was very weak. Second, the subordinate excitation peak beside the primary excitation peak of 465 nm (and 535 nm) was greatly enhanced, as shown in Figure 4e. This indicates that the crystal lattice orderly arrangement can enhance the energy level (^5^D_1_ and ^5^D_2_ of Eu^3+^) splitting via the enhanced crystal field effect. The enhancement of the subordinate excitation peak directly leads to broadening of the primary excitation peak, which is very important for phosphor materials with narrow excitation peaks in the applications of LD or LED as the excitation source.

### 2.6. Luminescence Quantum Efficiency

High quantum efficiency (QE) is very important for developing phosphor materials with applicable performance. In this study, the quantum efficiency was measured using a calibrated spectrometer and integrated sphere. The internal and external quantum efficiencies were calculated by Formulas (3) and (4):(3)$$\eta_{IQE}=\frac{\int R_{e}}{\int B_{s}-\int B_{r}}$$
(4)$$\eta_{EQE}=\frac{\int R_{e}}{\int B_{s}}$$
where $R_{e}$ is the emission spectrum, $B_{s}$ is the excitation light from the source, and $B_{r}$ is the excitation light with the sample in the integrating sphere. The measured IQE and EQE are summarized in Table 1. The measured IQE of ceramic samples was slightly lower than that of powder samples, and the IQE of the ceramic samples also showed a downward trend with the increase in calcination temperature. The decrease in grain boundary truly indicates the increase in surface defects on crystal grains was responsible, which is also a non-radiative channel. Therefore, the IQE was reduced in the ceramic samples in which the crystal grains had more surface defects compared with the grains in the powder sample. Furthermore, the EQEs of the ceramic samples were three times greater than that of the powder sample. As EQE is equal to the product of IQE and absorption efficiency, the absorption efficiency of the ceramic samples must have increased three-fold compared with that of the powder sample when the IQEs were the same. This is congruent with the results of the UV–Vis absorption spectra, as mentioned in Section 2.3. We also see that the EQEs of the ceramic samples were significantly large than those of the other Eu^3+^-doped red phosphor materials with a similar host lattice, as shown in Table 1.

### 2.7. Photoluminescence Decay Time

The luminescence decay curves of the powder and the ceramic samples are shown in Figure 5. The samples were excited at 465 nm and the emission at 616 nm was measured. These curves can be well-approximated by a second-order exponential fitting function. As per Formula (5) [36]:(5)$$I\left(t\right)=I_{0}+A_{1}\exp\left(\frac{-t}{\tau_{1}}\right)+A_{2}\exp\left(\frac{-t}{\tau_{2}}\right)$$
where I(t) refer to the PL intensity at time *t*, *I*_0_ is the baseline, $A_{1}$ and $A_{2}$ are the pre-exponential factors of each decay component, and *τ*_1_ and *τ*_2_ are decay times of each component. The average emission decay time ($\tau_{ave}$) can be calculated using Formula (6) [36],
(6)$$\tau_{ave}=\frac{A_{1}\tau_{1}{}^{2}+A_{2}\tau_{2}{}^{2}}{A_{1}\tau_{1}+A_{2}\tau_{2}}$$

The average emission decay time $\tau_{ave}$ shown in Figure 5, was calculated to be 0.716 for the powder and 0.646, 0.634, and 0.63 msec for the ceramic samples 900C, 1000C, and 1100C, respectively. The emission decay times of all ceramic samples were very similar and slightly lower than that of the powder sample. This suggests that, in the ceramic samples, the electronic relaxation time from the split ^5^D_2_ energy levels to the lowest transition energy level ^5^D_0_ was reduced.

### 2.8. Thermal Quenching Behaviour

The thermal stability of phosphor is one of the most important technological parameters in lighting and display technology [37,38], especially in high-power lighting devices. The thermal quenching behaviors of the ceramic and powder samples are presented in Figure 6a,b, respectively. The emission spectra were measured under 465-nm excitation light, and the sample working temperature was increased from room temperature to 300 °C in steps of 50 °C. The highest emission peak value at 616 nm and emission integral value over the range of 550 nm to 750 nm versus temperature are shown in Figure 6c,d for the ceramic and the powder samples, respectively.

The emission intensity of the powder sample started to decrease when the working temperature was higher than 75 °C, reaching 50% of the initial value at a working temperature of 200 °C; the emission was completely quenched at the working temperature of 300 °C. The emission intensity of the ceramic sample showed no degradation when the working temperature was up to 200 °C, and it was even larger than that at room temperature.

Thermal quenching mechanisms are usually different for each type of luminescence center, including a cascaded multi-phonon relaxation, thermal-dependent energy transfer (involving CTB), and crossover relaxation. In considering the large energy gap between the lowest excited state (^5^D_0_) and the highest ground state (^7^F_6_), the cascaded multi-phonon relaxation transition hardly occurred because more than 10 phonons are required to bridge the gap [39]. The crossover relaxation always appears with a changed photoluminescence decay time. Thus, the crossover relaxation intensity can be roughly estimated by the changed value of the photoluminescence decay time. In order to clarify the physical mechanism of thermal quenching, the photoluminescence decay curves of the powder and the ceramic samples were measured at different temperatures, increasing from room temperature to 300 °C (excited under 465 nm and recorded at 616 nm), as shown in Figure 7. The emission decay curves of both the powder sample and the ceramic samples did not depend on measurement temperatures, which indicates that the photoluminescence decay time basically does not change with changing temperature. Thus, the crossover relaxation contributes little to the thermal quenching behavior. Therefore, we confirmed that the thermal-dependent energy transfer (involving CTB) governed the thermal quenching behavior [40], which is described as the excited electrons jumping into the charge transfer band (CTB) after the absorption of thermal energy from the environment, and returning to the ground state through CTB via a non-radiative transition (as shown in Figure 8b).

The thermal energy absorbed from the environment to excite the electrons jumping into CTB that can cause emission quenching is called the activation energy (Δ*E*) of the thermal quenching process. To investigate the characteristics of thermal quenching behavior, the Δ*E* could be experimentally determined by using the Arrhenius equation in Formula (7) [41,42,43].
(7)$$\frac{I_{0}}{I_{T}}=1+A*EXP\left(-\frac{\Delta E}{kT}\right)$$
where $I_{T}$ is the emission intensity at the current temperature T, $I_{0}$ is the emission intensity at room temperature, *A* is a constant, and *k* is Boltzmann’s constant. Figure 8a represents the plots of ln[($I_{0}$/$I_{T}$) − 1] versus 1/(kT), and the value of the activation energy (ΔE) could be obtained from the slope of the linear fitting curve, which was 0.56 eV for the powder sample and 1.03 eV for the ceramic samples. The activation energy of the powder sample was much higher than that of Eu^3+^:Ca_19_Mg_2_(PO_4_)_14_ (ΔE = 0.14 eV) [44], Eu^3+^:Na_2_Tb_0_._5_(MoO_4_)(PO_4_) (ΔE = 0.2384 eV) [45], and Eu^3+^:NaSrLa(MoO_4_)_3_ (ΔE = 0.3461 eV) [34] and similar to that of the Eu^3+^:CaSnO_3_ (ΔE = 0.57 eV) sample [40].

The configuration coordinate diagram of energy levels and the thermal quenching process are shown in Figure 8b. Normally, after the electrons are excited, they relax from ^5^D_2_ to ^5^D_0_ (non-radiative transition) and jump to the ground state (radiative transition) accompanied by red photons being emitted. This process dominates at low temperatures—below 75 °C for the powder and below 200 °C for the ceramic samples. If the temperature increase is sufficient to provide the thermal activation energy Δ*E*, the excited electron will jump to the CTB band from ^5^D_2_, then transition to the ground state by a non-radiative method; taking a ceramic sample as an example, this is illustrated by the green arrow in Figure 8b [46]. The model successfully explains the process of thermal quenching with thermal activation energy.

## 3. Materials and Methods

### 3.1. Synthesis

The Eu^3+^:Y_2_(MoO_4_)_3_ powder sample was synthesized with an Eu^3+^-doping concentration of 50% using a combination of the sol–gel process and high-temperature solid-state reaction method [47,48]. During the synthesis process, Eu^3+^ ions will replace 50% of the Y^3+^ ions in the host material, Y_2_(MoO_4_)_3_, and form a EuY(MoO_4_)_3_ red phosphor material. All the reagents used in this work were of analytical-grade purity and were obtained from Sigma-Aldrich. Citric acid was used in the sol–gel process as the chelating agent and the molar dosage was equal to the molar dosage of cations in the solution. We add yttrium (III) nitrate, ammonium molybdate, and europium (III) nitrate into the citric acid solution sequentially to form a stoichiometric solution. Then, we heated the mixed solution to 100 °C for 12 h under continuous stirring with a magnetic stirrer with a 500 rad/min rotation. Thereafter, we evaporated water to form the gel and dried the gel at 300 °C for 2 h to obtain the precursor powder. Finally, the precursor powder underwent a solid-state reaction at a high temperature of 800 °C to obtain the powder that was subsequently pressed into powder cakes for sintering at high temperatures of 900 °C, 1000 °C, and 1100 °C, respectively. We investigated a wider range of calcination temperatures than those mentioned above; however, unfortunately, we did not achieve scientifically valuable data beyond 900 °C to 1100 °C. When the temperature was below 900 °C, the grain growth rate was very slow, and it became difficult for a dense ceramic sheet to form at the same time; when the temperature was above 1100 °C, the ceramic sheets had too many cracks. We marked the ceramic samples as 900C, 1000C, and 1100C, according to the different processing temperatures.

### 3.2. Characterization

The morphology and composition of the samples were studied using scanning electron microscopy (SEM) and energy-dispersive X-ray spectroscopy (EDS). The SEM image and EDS spectra were obtained using a Hitachi SU3500. Crystal structures were identified by X-ray diffraction (XRD) analysis. The XRD patterns were obtained using an Equinox1000 Sn.1612EQ1000137 diffractometer (Thermo Fisher; Horten, Norway) with Cu Kα radiation (λ = 1.5418 Å). The optical absorption spectrum was measured using a UV-2600 photo spectrometer (Shimadzu; Horten, Norway). The emission, excitation spectra, and electronic lifetime were measured using an Edinburgh FS05 Fluorescence Spectrometer. The quantum efficiency was measured with a calibrated AvaSpec-ULS2048-EVO PL spectrometer and AvaSphere-50 integrating sphere. In the thermal quenching experiment, samples were heated over a HT24S-24W metal ceramic heater (ThorLabs) and temperature-dependent photoluminescence was measured using the Edinburgh FS05 fluorescence spectrometer.

## 4. Conclusions

In this study, a new method for improving the photoluminescence properties of Eu^3+^:Y_2_(MoO_4_)_3_ red-emitting phosphor via crystal lattice orderly arrangement was successfully developed. In the powder samples, small crystal grains agglomerated with each other at high calcinating temperature to form ceramic samples with larger crystal grains. Comprehensive comparison of the Eu^3+^:Y_2_(MoO_4_)_3_ photoluminescence properties of ceramic and powder samples was presented. The ordered lattice arrangement in a larger range can significantly improve the luminescence properties of Eu^3+^:Y_2_(MoO_4_)_3_ samples. Specifically, compared with the powder sample, the ceramic samples had a 465-nm excitation peak that was broadened more than two-fold, e.g., from 2 nm to 4~6 nm. In addition, the absorption efficiency of the ceramic samples increased to 67% from 22% (for the powder sample) and the zero-quenching temperature increased to 200 °C from 75 °C (for the powder sample). Furthermore, the photoluminescence decay time of the ceramic samples also reduced slightly to 0.634 msec from 0.716 msec (for the powder sample), which will be helpful for improving the saturation power of the phosphor material. The Eu^3+^:Y_2_(MoO_4_)_3_ ceramic red phosphor with a long range of crystal lattice orderly arrangement is a potential candidate for use in lighting and display technology due to its excellent luminescence performance.

## Figures and Tables

**Figure 1 molecules-28-01014-f001:**
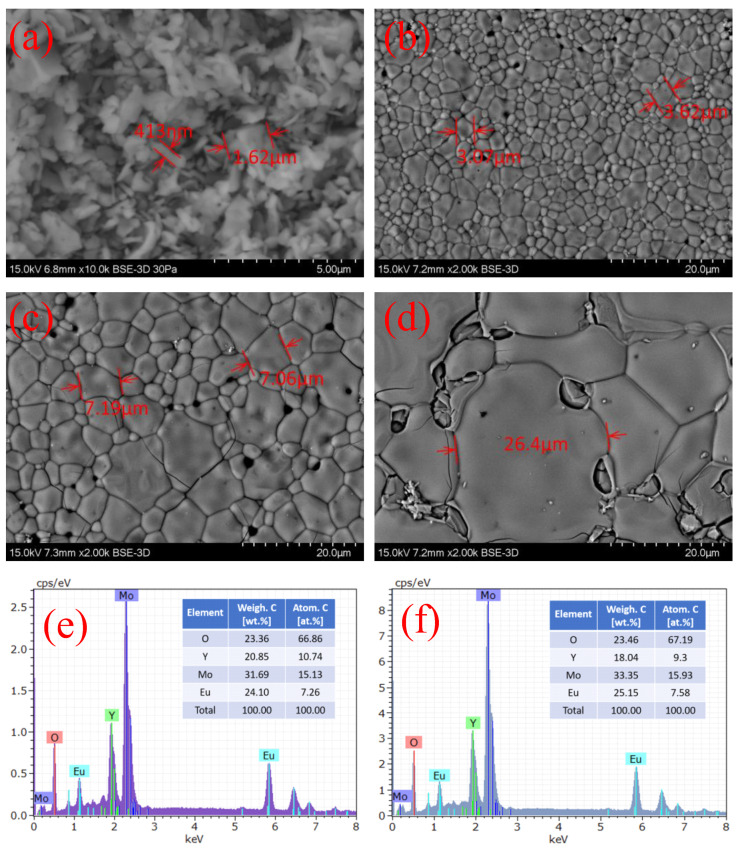
(**a**) SEM images of powder, (**b**) 900C, (**c**) 1000C, and (**d**) 1100C samples; (**e**) EDS spectra of powder and (**f**) ceramic samples. EDS results of three ceramic samples were the same, considering the measurement error; therefore, we used the results from 1000C as being representative for all ceramic samples.

**Figure 2 molecules-28-01014-f002:**
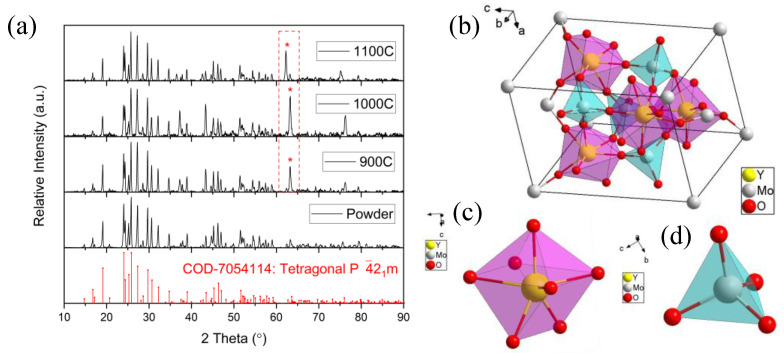
(**a**) XRD diffraction patterns of the powder and ceramic samples, compared with tetragonal crystal structure Y_2_(MoO_4_)_3_ standard card (COD-7054114); * point out the enhancement of the same diffraction peak; (**b**) unit cell tetragonal crystal structure of Y_2_(MoO_4_)_3_; (**c**) structure of single [YO_7_] group; (**d**) structure of single [MO_4_] group.

**Figure 3 molecules-28-01014-f003:**
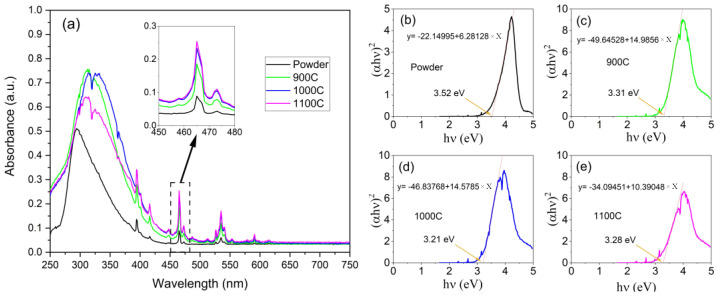
(**a**) UV−visible absorption spectra of four samples; inset of (**a**), UV−visible absorption spectra zoom at 465 nm; (**b**–**e**), plot of ${\left(\alpha hv\right)}^{2}$ versus hv and linear fitting results near the band edge, corresponding to the powder, 900C, 1000C, and 1100C samples, respectively.

**Figure 4 molecules-28-01014-f004:**
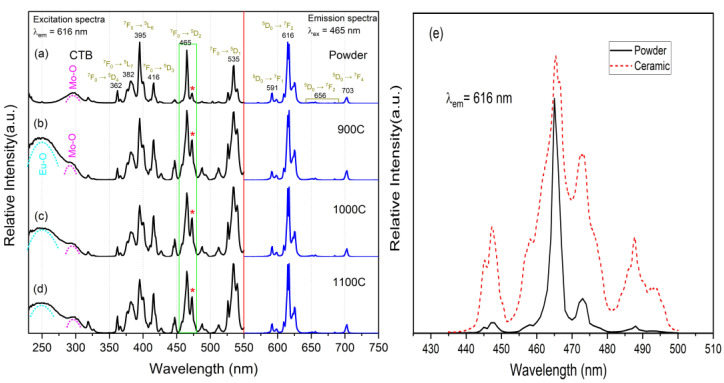
(**a**–**d**) Excitation and emission spectra of four samples, corresponding to powder, 900C, 1000C, and 1100C, respectively; (**e**) excitation spectra of powder and ceramic samples zoomed in to 465 nm; * point out the enhancement of the same excitation peak.

**Figure 5 molecules-28-01014-f005:**
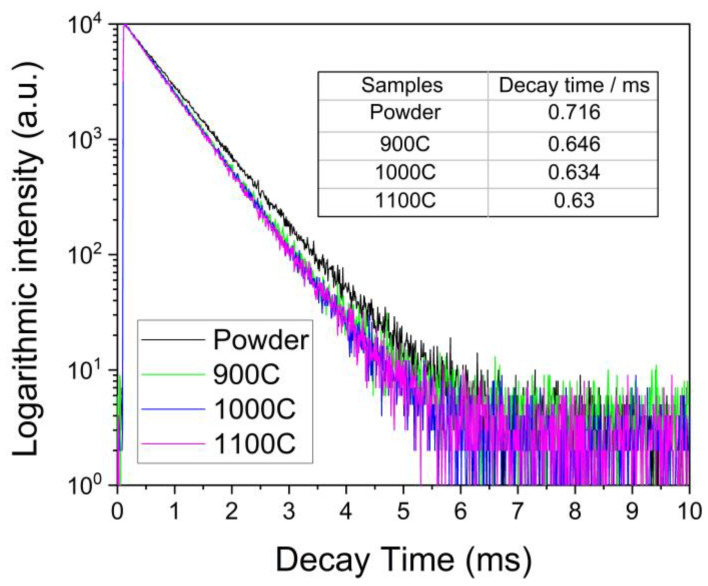
Photoluminescence decay curve of four samples with excitation and emission wavelength of 465 nm and 616 nm, respectively.

**Figure 6 molecules-28-01014-f006:**
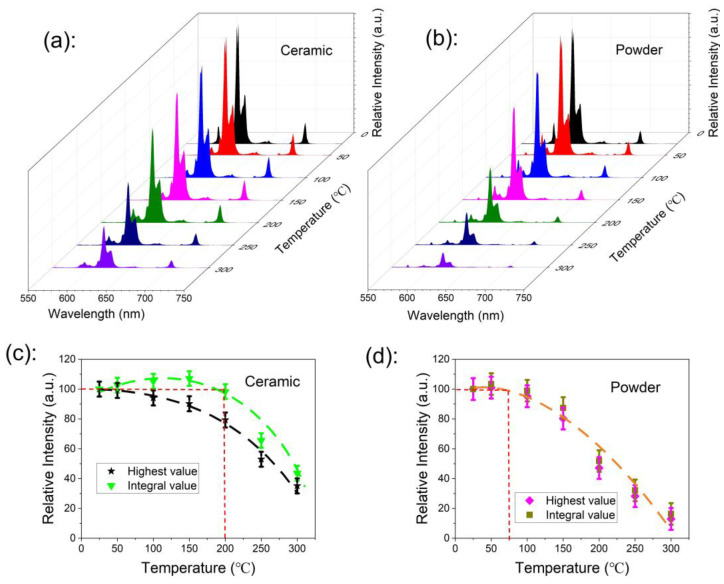
(**a**,**b**) The emission spectra thermal quenching behavior of ceramic and powder samples recorded under 465 nm, with a temperature increase from room temperature to 300 °C; (**c**,**d**) the highest emission peak value at 616 nm and emission integral value over the range of 550 nm to 750 nm versus temperature for the ceramic and powder samples.

**Figure 7 molecules-28-01014-f007:**
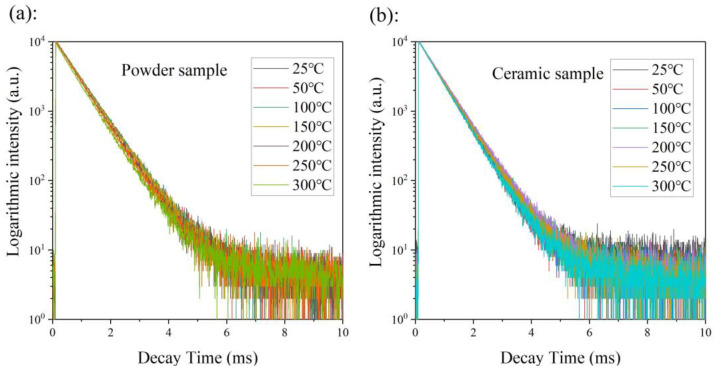
The photoluminescence decay curves of powder (**a**) and ceramic (**b**) samples were measured at temperatures increasing from room temperature to 300 °C (excited under 465 nm and recorded at 616 nm).

**Figure 8 molecules-28-01014-f008:**
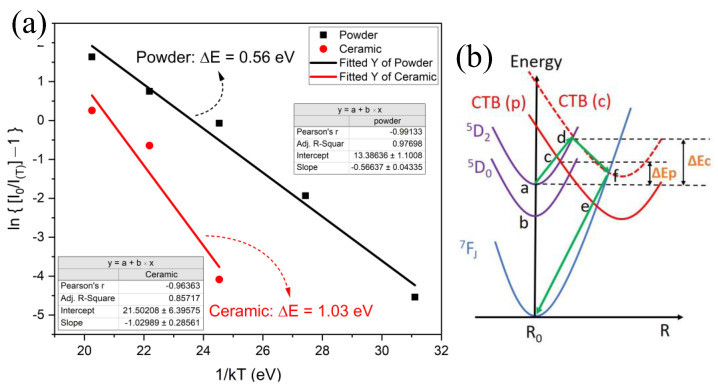
(**a**) The linearly fitted activation energy for the thermal quenching process of powder and ceramic samples; (**b**) configurational coordinate diagram to show the thermal quenching mechanism and pathway for P (powder) and C (ceramic) samples.

**Table 1 molecules-28-01014-t001:** The internal quantum efficiency (*IQE*), external quantum efficiency (*EQE*), and absorption efficiency (Abs) of our samples and reported samples, as a comparison.

Samples	IQE	EQE	Abs	Ref.
Eu^3+^:Y_2_(MoO_4_)_3_ Powder	0.95	0.21	0.22	This work
Eu^3+^:Y_2_(MoO_4_)_3_ 900C	0.93	0.51	0.55	This work
Eu^3+^:Y_2_(MoO_4_)_3_ 1000C	0.91	0.61	0.67	This work
Eu^3+^:Y_2_(MoO_4_)_3_ 1100C	0.88	0.57	0.65	This work
Eu^3+^/Au:Y_2_(MoO_4_)_3_	0.92	0.10	0.11	[20]
Eu^3+^:Y_2_(MoO_4_)_3_	-	0.2238	-	[19]
Eu^3+^:Na_2_Gd(PO_4_)(MoO_4_)	0.90	0.37	0.41	[35]

## Data Availability

The more research data are available from the authors on request.

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
