# Peer review of "Improved Photoluminescence Performance of Eu3+-Doped Y2(MoO4)3 Red-Emitting Phosphor via Orderly Arrangement of the Crystal Lattice"

_molecules, 2023, doi:10.3390/molecules28031014_

Round 1
Reviewer 1 Report
Chen et. al. reported a normal strategy of thermal treatment to improve the PL performance of Eu: Y2(MoO4)3 and revealed the underlying photophysics. Although the strategy is widely reported, the results are systematical. Major revision is needed before further consideration.
Some problems are listed below:
Q1: Since calcination can help the order arrangment of crystal lattice, a wider range of calcination temerature should be investigated and compared.
Q2: The authors attributed to the improved PL performance to the order arrangement of lattice. However, the order arrangement is hard to evaluated or quantified. The decrease of grain boundary also indicates the increase of surface defects, which is also a non-radiative channel.
Q3: The authors state that the diffraction peaks at 62-64 result from the ordered lattice, this must be supported by other evidence and literature.
Q4: Actually, EDS cannot tell us the impurities with a small content.
Q5: As an ion emission center, the calculation of bandgap is meaningless.
Q6: Usually, the order arrangement of lattice will result in weaker crystal field effect and smaller strain, thereby generating narrower emission and PLE. What’s the mechanism for the different results in this case?
Author Response
Thank you very much for your suggestions that help us to improve the quality of the paper. In the attachment, you see the details about our changes according to your comments.
Please see the attachment.

Reviewer 2 Report
In this research, the authors achieved lattice ordering by the powder aggregation and diffusion at high temperature. Compared with Eu3+:Y2(MoO4)3 powder, full width at half maximum (FWHM) of the excitation peak of the ceramic is broaden by 2 to 3 times. More interestingly, Eu3+:Y2(MoO4)3 ceramic material showed little thermal quenching below temperature of 473 K making it useful for high lumen light output operating at high temperature. I considered it can be published after a minor revision:
(1) I think it would be better to have the key words in a lattice order. The unit of ℃ in the text is wrong, and there is no space between the number and the letter.
(2) The SEM description of Figure 1 mentioned the change of particle size, I think the author had better mark it on the Figure 1.
(3) Corresponding data should be added to illustrate the phenomenon of grain boundary reduction and grain size increase. The author should explain the reasons for choosing 900℃, 1000℃ and 1100℃ in the article.
(4) The quality of the pictures in this paper needs to be improved, and the author should refer to more literature to enrich the article. add some references such as “Nanomaterials, 2022, 12(3): 446., Scientific Reports, 2021,11(1):2495. International Journal of Heat and Mass Transfer, 2018, 120:1-8.”
(5) Errors in the manuscript need to be corrected, and pay attention to improving the grammar and the quality of pictures in the manuscript.
Author Response

(The authors gave the same response as above.)

Reviewer 3 Report
Chen et al have synthesized Eu doped Y2(MoO4)3 at 800, 900, 1000 and 1000 Deg C and compared the their optical, and photoluminescence properties at room temperature and at high temperatures. I recomend for major revision
The following concerns
1. The manuscripts need significant improvements in term of interpreting the results well as writing errors.
2. As the crystallinity increase, the emission, decay time should increase. Higher ordering in crystal should lead shaper peaks and higher decay time. Authors have observed that the ceramic samples show broader peak compared to powder samples and lower lifetimes that the contracting the claims. On the same ground, the IQE should increase.
3. As temperature increase, the non-radiative transition increase, thus emission intensity and decay values should decrease. Authors observed that emission intensity decrease but the decay values remain the same for powder the ceramic samples.
4. Authors should mention the concentration of the Eu ions in the synthesis section.
5. Authors need to compare the undoped and Eu-doped sample to confirm Eu-O and Mo-O absorption/ charge transfer peak.
6. In Figure 4e, its looks like that there is phase separation and authors need to analyze more representative samples with different Eu concentration.
Author Response

(The authors gave the same response as above.)

Round 2
Reviewer 1 Report
Accepted as it is
Author Response
Thank you very much.
Reviewer 3 Report
Authors have addressed the comments, but one of the comments have raised further questions.
1. Authors said the concentration Eu3+ ions is 50 % where as EDS shows around 7.2-7.6 %. This looks to vague.
2. If the concentration is 50 %, there should be solubility of Eu3+ ions in Y2(MoO4)3 needs to be tested with varying concentration say 10% to 50%.
3. Minor correction on page 5, 'xcitation' should be corrected to 'Excitation'
Author Response
Thanks very much for your suggestions that help us to improve the quality of the paper. In the following, you see the details about our changes according to your comments.
1,Authors said the concentration Eu3+ ions is 50% where as EDS shows around 7.2-7.6%. This looks to vague.
We say that the doping concentration of Eu3+ ions is 50%, which means that Eu3+ ions replace 50% of Y3+ ions in the host material Y2(MoO4)3.
EDS shows that the Eu content is 7.2-7.6%, and the Y content is 10.7-9.3%. The doping concentration of Eu3+ is calculated to be 40-45%, which is close to the design value -50%.
To reduce the confusion, we have added the following text in the revised version:
During synthesis process, Eu3+ ions will replace 50% of the Y3+ ions in the host material Y2(MoO4)3, and form EuY(MoO4)3 red phosphor material.
3. Minor correction on page 5, 'xcitation' should be corrected to 'Excitation'
Thank you very much for pointing out this error, which has been corrected in revised version.